# From Social Networking Site Use to Subjective Well-Being: The Interpersonal and Intrapersonal Mediating Pathways of Prosocial Behavior among Vocational College Students in China

**DOI:** 10.3390/ijerph20010100

**Published:** 2022-12-21

**Authors:** Bryant Pui Hung Hui, Algae Kit Yee Au, Jacky Chi Kit Ng, Xinmiao Song

**Affiliations:** 1Department of Applied Social Sciences, Faculty of Health and Social Sciences, Hong Kong Polytechnic University, Hong Kong SAR, China; 2Department of Sociology, Faculty of Social Sciences, University of Hong Kong, Hong Kong SAR, China

**Keywords:** social networking site use, well-being, prosocial behavior, vocational college, China

## Abstract

In view of the growing importance of social networking sites (SNS) to adolescents and the mixed and inconclusive empirical evidence on the relationships between SNS use and their well-being, the present study aimed to investigate the associations of social function use intensity (SFUI) and entertainment function use intensity (EFUI) with adolescent life satisfaction and self-esteem, and examine the mediating roles that general prosocial behavior and school volunteering may play in the links. Drawing from the findings of a self-administered online survey with a valid sample of 3452 adolescents (mean age = 18.21) from 10 vocational colleges across four regions of China, our results demonstrated that there was an indirect positive effect of SFUI on adolescent life satisfaction and self-esteem via two interpersonal pathways of general prosocial behavior and school volunteering. We also discovered that there was an indirect negative effect of EFUI on adolescent life satisfaction and self-esteem via an intrapersonal pathway of school volunteering. Our findings provided empirical support for the differential effects of SFUI and EFUI on adolescent life satisfaction and self-esteem through the interpersonal and intrapersonal pathways, and unpacked the mediating roles of general prosocial behavior and school volunteering in these mechanisms.

## 1. Popularity and Influence of SNS among Adolescents

Social networking sites (SNS), referring to the highly interactive online platforms for sharing, co-creating, and modifying user-generated content [1], have become an influential and integral part of people’s daily routines. In 2021, more than half of the world’s population used SNS and spent an average of nearly 2.5 hours daily on these sites [2]. The importance of SNS is particularly salient among adolescents, as this unique demographic cohort was born into a digital world, where they gain intense exposure to interactive technology and become the main users of SNS [3,4]. Not surprisingly, SNS use is also common among Chinese adolescents. By the end of 2021, the number of young Internet users (aged 10–19) in China had reached 137 million, and most of them use instant messaging and SNS on mobile phones [5]. With the rapid expansion of SNS functions and the rise of new platforms, young people can now use SNS for diverse purposes. Besides traditional social activities, such as keeping in touch with family and friends, sharing opinions, discussing with others, or posting updates, they can also undertake other non-social activities such as watching livestreams, shopping, and gaming on SNS [2,5]. The use of SNS among adolescents is evident, yet the association with their psychological indicators is a polarized and much-debated research subject [6,7], and the majority of research focuses on the Western context [8,9,10,11,12]. In view of the large population of young SNS users in China and the unique nature of the Chinese SNS ecosystem characterized by homegrown platforms [2], this study paid special attention to the psychological indicators associated with SNS use among Chinese adolescents.

## 2. SNS Use and Adolescent Well-Being

One emerging line of research seeks to investigate the relationship between SNS use and adolescent well-being [3,13,14,15], not just because well-being is a key psychological construct, but also because adolescence is a crucial developmental stage for life-long well-being [16] and adolescent well-being can bring encompassing personal and societal benefits [17]. Well-being refers to one’s global evaluation of life [18]. It is widely regarded as a basic human goal [19] and is empirically found to be a predictor of various desirable life outcomes [20,21]. There are two contrasting views regarding the relationships between SNS use and well-being. The positive view claims that SNS use creates social capital and elicits feelings of social connectedness, thus enhancing well-being. On the contrary, the negative view posits that SNS use provokes social comparison and envy, thereby undermining well-being [22,23]. Both views have received considerable empirical support [24]. Meanwhile, there were also mixed and inconsistent research findings on the associations of SNS use with adolescent well-being. For example, some studies revealed that the use of online communication was at least partially accountable for the decrease in adolescent well-being in terms of life satisfaction and self-esteem [25], whereas other studies demonstrated that SNS use did not have substantial associations with adolescent well-being in the long run [24]. These inconclusive findings indicate that more research efforts should be devoted to understanding the links between SNS use and adolescent well-being. In light of the above, this paper focused on the associations between SNS use and two key indicators of adolescent well-being, namely life satisfaction and self-esteem.

### 2.1. SNS Use and Adolescent Life Satisfaction

Life satisfaction refers to one’s overall assessment of life quality and is a key indicator of well-being [26]. Though it is mostly regarded as intrapersonal contentment [26], the construct is also partly accounted for by interpersonal factors such as one’s social experience [27]. As a convenient tool for gaining social experience, SNS may help strengthen one’s interpersonal relationships and enhance supportive interaction, leading to increased life satisfaction [28,29]. However, unwanted or unnecessary social comparisons against one’s life may also emerge as the side-products of the expanded online social circles, resulting in decreased life satisfaction [29,30]. Although SNS is a major source of adolescents’ social life, its impacts on their life satisfaction have not been determined conclusively. In particular, positive associations of SNS use with adolescent life satisfaction have only been found in a limited number of studies [31,32], while its overall negative associations across the adolescent population have been observed in a large-scale nationally representative panel data analysis—though the associations are just small, nuanced, and inconsistent [10].

### 2.2. SNS Use and Adolescent Self-Esteem

Self-esteem refers to the subjective evaluation of one’s self-worth [33,34] and it is largely derived from social acceptance and approval [35]. According to some studies, the use of SNS can enhance young people’s self-esteem by helping them maintain and strengthen peer relationships and get positive feedback [36]. For instance, a study in Spain found that socializing on Tuenti (a popular SNS for Spanish adolescents) positively predicted self-esteem among adolescents [36]. However, some researchers assert that constant exposure to idealized depictions of others on SNS may make adolescents feel inadequate and undermine their self-esteem [37]. For example, SNS use is found to be negatively associated with self-esteem among Scottish adolescents [9] and US first-year college students [8]. A more recent meta-analysis [38] also revealed a small and negative association between SNS use and self-esteem. Nonetheless, these negative relationships were found to be more pronounced for problematic SNS use, indicating that general measures might not be able to capture the diversity of SNS activities.

## 3. Two Lines of Inquiry

The intriguing and inconsistent research findings regarding the associations between SNS use and adolescent well-being have called for more nuanced investigations, and two lines of inquiry are deemed to be particularly pertinent. The first line focuses on identifying different SNS activities and examining their possible differential associations with adolescent well-being. Its objective is to address the methodological problems associated with assessing SNS usage as a whole while overlooking the effects of different forms of SNS use [39]. On the other hand, the second line aims at investigating the underlying mechanism between SNS use and adolescent well-being by including potential mediators in the examination [3]. The current research was guided by both lines of inquiry.

### 3.1. Types of SNS Use

One predominant categorization of SNS uses is based on their *forms*, which separates them into *active* vs. *passive* uses [11,23,40]. Active SNS usage involves direct exchanges with others, and typical examples include direct communication, broadcasting, posting status updates, sharing links, and sending messages [23,40]. Passive usage refers to SNS activities that aim to monitor others’ lives without direct exchanges, and common examples include scrolling through news feeds and reading others’ profiles, pictures, or status updates [23]. The differential associations of these two types of SNS use with well-being have been largely supported by empirical evidence, which suggests that active use enhances well-being and passive use deflates well-being [23]. However, the active vs. passive SNS use dichotomy has been seriously challenged from the theoretical and empirical viewpoints [15]. Theoretically, passive SNS use such as browsing information involves the active cognitive process of information selection, processing, and interpretation [15,41]. Empirically, analyses of an experience-sampling dataset showed that the hypothesized positive association of active SNS use and the negative association of passive SNS use with well-being only hold true for a negligible proportion of adolescents. For the majority, no consistent patterns have been observed [42,43].

Another classification of SNS uses focuses on their *functions*, which divides them according to their *social* vs. *entertainment* purposes. This classification was proposed by Li and colleagues [44], who developed and validated the Social Networking Activity Intensity Scale (SNAIS) for junior middle school students in China to reflect two distinct behavioral patterns of SNS use, viz., *Social Function Use Intensity (SFUI)* and *Entertainment Function Use Intensity (EFUI)*. SFUI refers to the frequency of using the social functions of SNS, including sending messages, chatting, making or replying to comments, sharing or forwarding content, posting, updating self-status, and editing profiles [44]. Given that SFUI focuses on building up or maintaining social relationships, it can be conceptualized as the *interpersonal* aspect of SNS use. EFUI refers to the frequency of using the entertainment functions of SNS, including surfing entertainment or news, watching videos or listening to music, playing games or applications, and buying or giving virtual goods [44]. Given that EFUI mainly concerns personal enjoyment, it can be conceptualized as the *intrapersonal* aspect of SNS use. Compared with the active vs. passive categorization, the SFUI vs. EFUI classification can better reflect the rising use of SNS for entertainment today. Indeed, a recent global survey revealed that finding funny/entertainment content is the top reason for using emerging SNS including TikTok and Reddit, and the second-largest reason for using other popular SNS including Twitter, Snapchat, and Pinterest [2]. Taken together, SFUI vs. EFUI appears to be a more desirable classification to depict adolescents’ behavioral patterns of SNS use. Thus, the possible differential associations of SFUI and EFUI with adolescent well-being warrant further investigation.

### 3.2. Prosocial Behavior: A Possible Mediator

One potential mediator between SNS use and adolescent well-being is prosocial behavior, which refers to volitional acts that aim to benefit others [45,46,47]. An example of prosocial behavior is volunteering, which is a formal helping act planned and executed in the organizational context for the betterment of the community or to the benefits of a specific group in need [46,47]. In China, youth volunteering activities are generally funded and initiated by the government and organized by schools [48]. Being one of the most powerful socialization agents, schools can exert profound influences on adolescents’ life and prosocial involvement [49]. Therefore, we reasoned that school volunteering is a distinctive prosocial behavior during adolescence. Apart from that, young people can also engage in other prosocial behavior beyond the school context, such as offering unplanned and unorganized daily private assistance with no specific target beneficiaries. For instance, they can give directions to a stranger or help an acquaintance move house [45,46,47,50,51]. There are two main reasons why adolescent prosocial behavior is unique. First, adolescence is an important life phase characterized by the rapid development of the cognitive, physical, and social ability required to perform prosocial behavior, making it different from childhood and adulthood. Second, adolescence is a critical stage of identity formation, and prosocial qualities ingrained in self-identity during this period are likely to impact prosocial behavior for life [52]. Extant research has examined the antecedents and consequences of adolescent prosocial behavior, including how SNS use impacts prosocial behavior, and how prosocial behavior influences well-being. The present study seeks to take previous works forward.

***SNS Use and Prosociality.*** Whether SNS use promotes or hinders adolescent prosocial behavior has been an ongoing debate between two contrasting perspectives [53]. The optimistic view suggests that impression management on SNS can be extended to real life through prosocial behavior, as prosociality signals positive human qualities that contribute to social approval and improve one’s social standing [54,55]. It is also believed that as the primary purpose of SNS is to establish and strengthen social relationships, SNS activities can enhance social connection, leading to civic engagement and prosocial behavior [53,56]. Some even argue that SNS is the most desirable approach to engage young people in prosocial charitable causes [57]. In contrast, the pessimistic view proposes that the indiscriminate one-to-many communication on SNS ignores the diverse interests of recipients and induces a sense of egocentrism, resulting in reduced prosocial behavior [58]. This pessimistic claim has gained some empirical support from experimental studies among Facebook users, which found that participants behaved more selfishly in dictator games and put in less effort in the volunteered data coding task after they published a wall post [58]. Nonetheless, these findings were all based on adult samples, and whether the same negative associations of SNS use can be found in adolescents, awaits further investigation. Besides, a recent longitudinal analysis found no support for the enduring associations of SNS use with adolescent prosocial behavior [59]. Previous research also did not consider different forms of SNS use and examine their differential associations with distinctive types of prosocial behavior. In short, the relationship between SNS use and adolescent prosocial behavior remains unclear thus far.

***Prosocial Behavior and Well-Being.*** Compared with the relationship between SNS use and prosocial behavior, the positive link between prosocial behavior and well-being appears to be less contestable. As reported by a recent large meta-analysis (*K =* 201), a wide range of prosocial behavior is positively associated with well-being, though the effect size of informal prosocial behavior is relatively larger than that of formal prosocial behavior [45]. One prevailing mechanism posits that prosocial behavior increases self-evaluation and perceived competence, and distracts people from overfocusing on their own troubles and stress, which consequently enhances well-being [60]. The positive link between prosocial behavior and adolescent well-being has been well-documented as well [61,62,63]. For example, a longitudinal study found that prosocial behavior positively predicted adolescent life satisfaction two years later [62]. Similarly, another longitudinal study revealed that prosocial behavior positively predicted self-esteem during the transition from adolescence to young adulthood [63]. In brief, past literature widely supports the positive link between prosocial behavior and adolescent well-being.

## 4. Interpersonal and Intrapersonal Meditating Mechanisms of SNS Use

To delineate the differential associations of the two forms of SNS use with well-being among Chinese adolescents, we proposed that SFUI would positively predict life satisfaction and self-esteem through an *interpersonal* pathway, mediated by general prosocial behavior and school volunteering. In contrast, EFUI would negatively predict life satisfaction and self-esteem through an *intrapersonal* pathway, also mediated by general prosocial behavior and school volunteering.

### 4.1. Interpersonal Mediating Mechanism of SFUI

SFUI represents the *interpersonal* aspect of SNS use, as it covers activities that are all meant to initiate or sustain social interaction, such as chatting, commenting, and updating self-status [44]. These activities would facilitate social relationship development and increase the sense of social connectedness, possibly serving as catalysts for general prosocial behavior and school volunteering. Helping others would distract an individual from dwelling on one’s own problems, eventually improving life satisfaction [60]. Having the ability to help others would also strengthen an individual’s perceived competence and sense of self-worth, ultimately bolstering self-esteem [63]. In other words, SFUI would go through an interpersonal pathway mediated by both general prosocial behavior and school volunteering to positively predict life satisfaction and self-esteem.

### 4.2. Intrapersonal Mediating Mechanism of EFUI

In contrast, EFUI represents the *intrapersonal* aspect of SNS use, as it covers activities that aim for personal enjoyment, such as watching videos, listening to music, and playing games [44]. Simply indulging in one’s own pleasure without interacting with others may strengthen one’s sense of egocentrism and reduce one’s interest in others’ welfare, thus lowering one’s tendency to engage in any kind of helping acts [58]. While increased prosocial behavior improves well-being [64], decreased prosocial behavior may negatively predict life satisfaction by reducing the opportunity to distract oneself from personal troubles and stress through acting kindly [60]. Moreover, decreased prosocial behavior also deprives an individual of the chance to demonstrate the ability to help others, which would in turn lower one’s perceived competence and self-efficacy, ultimately undermining self-esteem [63]. Hence, EFUI would go through an intrapersonal pathway mediated by both general prosocial behavior and school volunteering to negatively predict life satisfaction and self-esteem.

## 5. The Present Study

The present research set out to make an important and original contribution to the scarce literature on SNS use, prosocial behavior, and well-being among Chinese adolescents—the key users in the largest SNS market in the world. We aimed to examine the associations of SNS use with adolescent well-being and unpack the underlying mechanism of such associations in our two objectives.

### 5.1. Differential Associations of SNS Use with Adolescents’ Well-Being

Our first objective was to investigate the differential associations of two types of SNS use with adolescent well-being. In view of the mixed and inconclusive empirical evidence on the topic, we adopted the SFUI and EFUI dichotomy to analyze the associations of SNS use with the life satisfaction and self-esteem of adolescents. Given that China is the largest SNS market in the world [65] and has a huge number of young SNS users, we conducted a large cross-sectional online survey on a student sample (*N* = 3452) from 10 vocational colleges in four regions of China. Although there has been an expansion of vocational education in China since the 1980s and about 40% of Chinese adolescents end up in vocational colleges [66], limited psychological research has been conducted for this “underprivileged” community, especially on their SNS use, prosocial behavior and well-being [67,68,69,70]. Based on the literature reviewed above, we expected that SFUI would positively predict adolescent life satisfaction and self-esteem via an interpersonal pathway, whereas EFUI would negatively predict adolescent life satisfaction and self-esteem via an intrapersonal pathway.

### 5.2. Mediation Effects of Prosocial Behavior

Our second objective was to explore the mediation effects of prosocial behavior in the SNS use and adolescent well-being linkages. To further examine the critical underlying mechanism, we sought to test the potential mediating roles that general prosocial behavior and school volunteering in the interpersonal path of SFUI and the intrapersonal path of EFUI to adolescent life satisfaction and self-esteem. We expected that SFUI would positively predict both general prosocial behavior and school volunteering, which in turn positively predict adolescent life satisfaction and self-esteem. We also expected that EFUI would negatively predict both general prosocial behavior and school volunteering, which then positively predict adolescent life satisfaction and self-esteem. The conceptual models of these two mediating pathways are illustrated in Figure 1.

## 6. Materials and Methods

### 6.1. Research Design

This study used a cross-sectional survey and a convenience sample of vocational college students in China. Principals and teachers from a number of vocational colleges agreed to join the study and facilitated data collection after they attended a conference for vocational school educators in Xi’an, China.

### 6.2. Participants

Participants were recruited from 10 vocational colleges in the Central (Hubei province), Eastern (Anhui, Guangdong, and Zhejiang provinces), Northern (Inner Mongolia province), and Western (Guizhou and Gansu provinces) regions of China.

### 6.3. Procedures

A total of around 8100 vocational college students were invited to take part in an online self-administered survey by teachers and principals of their colleges. After signing an online informed consent, 5021 participants completed the survey on a voluntary basis without monetary incentive (average response rate = 61.99%). To identify careless responses [71,72], we embedded six quality-check items (e.g., for this item, please select “*strongly agree*”). The final sample included 3452 participants (1766 males, 51.20%; *M*_age_ = 18.21, *SD*_age_ = 2.01) who answered four or more quality-check items correctly and finished the survey in 15 min or more (the median response time = 34 min). The present study was approved by the ethics committee of the affiliated university of the first author. The data collection was approved and assisted by the vocational college administration.

### 6.4. Measures

*SNS function use intensity.* The SNS function use was measured by the Social Networking Activity Intensity Scale (SNAIS) [44]. It consists of 14 items assessing two types of SNS function use intensity, which include 10 items on social function use intensity (SFUI) and 4 items on entertainment function use intensity (EFUI). Participants rated the frequency of performing online networking activities in the past month (e.g., “shared/forwarded content” (SFUI), “watched video/listened to music” (EFUI)) on a 4-point Likert scale ranging from 1 (*never*) to 4 (*always*). The Cronbach’s alphas of SFUI and EFUI in the present study were 0.92 and 0.72, respectively.

*General prosocial behavior.* The 20-item Self-Report Altruism Scale (SRAS) [50] was used to tap participants’ general prosocial behavior across a variety of categories (e.g., “I have given directions to a stranger”, “I have offered my seat on a bus or train to a stranger who was standing”, and “I have given money to a charity”). This served as a measure of general prosocial behavior, and participants rated the frequency of items on a 5-point Likert scale ranging from 1 (*never*) to 5 (*very often*). The Cronbach’s alpha in the present study was 0.91.

*School volunteering.* To complement the SRAS which does not cover school volunteering work, we added a single item “I have participated in school volunteering work”, so that participants could report the frequency by rating from 1 (*never*) to 5 (*very often*).

*Life satisfaction.* Participants were asked to complete the Satisfaction with Life Scale (SWLS) [26], a 5-item measure for the cognitive evaluation of one’s life in general on a 7-point Likert scale (1 = *strongly disagree* to 7 = *strongly agree*). A sample item is “I am satisfied with my life”. The Cronbach’s alpha in the present study was 0.90.

*Self-esteem.* The 10-item Rosenberg Self-Esteem Scale [34] was employed to measure one’s evaluation of self-worth. Items are anchored with a 4-point Likert scale ranging from 1 (*strongly disagree*) to 4 (*strongly agree*). Sample items include “At times I think I am no good at all”, and “I take a positive attitude toward myself”. The Cronbach’s alpha in the present study was 0.70, after removing the item “I wish I could have more respect for myself” with a negative item-total correlation.

*Smartphone use frequency.* Participants rated the frequency of smartphone use in the past month on a 5-point scale ranging from 1 (*never*) to 5 (*more than once per day*).

Six quality-check items and demographic information (e.g., age and gender) were also included.

### 6.5. Data Analysis

All statistical analyses were conducted using SPSS 25.0 (IBM Corp., Armonk, NY, USA). Pearson correlations were employed to examine the bivariate associations among all variables. To test the hypothesized models, mediation analysis was conducted to examine whether the relationship between SNS function uses (i.e., SFUI and EFUI) and well-being indicators (i.e., life satisfaction and self-esteem) would be mediated by general prosocial behavior and school volunteering. We used SPSS Macro for Multiple Mediation to simultaneously estimate the indirect effect of a predictor variable on an outcome variable through more than one mediating variables in a single model [73]. Bootstrapped mediation tests were conducted based on 5000 bootstrapped resamples.

## 7. Results

### 7.1. Descriptive Statistics

Table 1 illustrates the descriptive results of all variables of interest. We present the mean, standard deviation, and reliability coefficient of each variable, along with the Pearson correlation coefficient for each pair of variables. The correlational results showed that SFUI was positively correlated with general prosocial behavior, *r* = 0.27, *p* < 0.001, school volunteering, *r* = 0.20, *p* < 0.001, life satisfaction *r* = 0.25, *p* < 0.001, and self-esteem, *r* = 0.13, *p* < 0.001. EFUI was positively correlated with general prosocial behavior, *r* = 0.15, *p* < 0.001, school volunteering, *r* = 0.07, *p* < 0.001, life satisfaction, *r* = 0.20, *p* < 0.001, and self-esteem, *r* = 0.10, *p* < 0.001. General prosocial behavior was positively correlated with life satisfaction, *r* = 0.16, *p* < 0.001, and self-esteem, *r* = 0.16, *p* < 0.001. Similarly, school volunteering was positively correlated with life satisfaction, *r* = 0.13, *p* < 0.001, and self-esteem, *r* = 0.15, *p* < 0.001.

The positive correlations between EFUI and general prosocial behavior and school volunteering are probably due to a large shared association between SFUI and EFUI, *r* = 0.63, *p* < 0.001. Therefore, we also computed a partial correlation in which EFUI was negatively correlated with school volunteering, *r* = −0.07, *p* < 0.001, and general prosocial behavior, *r* = −0.03, *p* = 0.124—albeit not statistically significant—when SFUI was controlled for. Similarly, after controlling for EFUI, the partial correlation between SFUI was positively correlated with general prosocial behavior, *r* = 0.23, *p* < 0.001, and school volunteering, *r* = 0.19, *p* < 0.001.

The correlational results were basically in the same directions as our hypotheses and supported our further examination in the mediation analysis.

### 7.2. Mediation Analysis

We tested our hypothesized models through a series of mediation analyses. To account for the shared association between SFUI and EFUI (*r* = 0.63) in the mediation models being estimated, both SFUI and EFUI were included in all models. In addition, other covariates of sex, age, smartphone use frequency, and region of schools were included to prevent possible confounding and epiphenomenal associations due to covariates. All main model coefficients are presented in Figure 2.

#### 7.2.1. Analysis of the Effects of SFUI

In the models examining the indirect effects of SFUI on well-being indicators (Models 1a and 1b), SFUI significantly and positively predicted general prosocial behavior, *B* = 0.25, *p* < 0.001, which in turn positively predicted life satisfaction, *B* = 0.12, *p* < 0.001. Similarly, SFUI significantly and positively predicted school volunteering, *B* = 0.38, *p* < 0.001, which in turn positively predicted life satisfaction, *B* = 0.04, *p* = 0.031 (see Model 1a). Bootstrapping results revealed the significant indirect effect of SFUI on life satisfaction through both general prosocial behavior, *B* = 0.03, bias-corrected 95% CI [0.010, 0.052], and school volunteering, *B* = 0.02, bias-corrected 95% CI [0.001, 0.035]. The overall model *R*^2^ was significant, *F* = 36.83, *p* < 0.001, explaining 8% of the variance in life satisfaction. Consistently shown in Model 1b, SFUI significantly and positively predicted general prosocial behavior, *B* = 0.25, *p* < 0.001, which in turn positively predicted self-esteem, *B* = 0.05, *p* < 0.001. Similarly, SFUI significantly and positively predicted school volunteering, *B* = 0.38, *p* < 0.001, which in turn positively predicted self-esteem, *B* = 0.02, *p* < 0.001. Bootstrapping results revealed the significant indirect effect of SFUI on self-esteem through both general prosocial behavior, *B* = 0.02, bias-corrected 95% CI [0.005, 0.019], and school volunteering, *B* = 0.01, bias-corrected 95% CI [0.004, 0.016]. The overall model *R*^2^ was significant, *F* = 25.50, *p* < 0.001, explaining 6% of the variance in self-esteem.

#### 7.2.2. Analysis of the Effects of EFUI

In the models examining the indirect effects of EFUI on well-being indicators (Models 2a and 2b), EFUI did not predict general prosocial behavior, *B* = 0.00, *p* = 0.987, but significantly and negatively predicted school volunteering, *B* = −0.08, *p* = 0.035, which in turn positively predicted life satisfaction, *B* = 0.04, *p* = 0.031 (see Model 2a). Bootstrapping results revealed the significant indirect effect of EFUI on life satisfaction through school volunteering, *B* = −0.00, bias-corrected 95% CI [−0.010, −0.000]. The overall model *R*^2^ was significant, *F* = 36.83, *p* < 0.001, explaining 8% of the variance in life satisfaction. Likewise, as shown in Model 2b, EFUI did not predict general prosocial behavior, *B* = 0.00, *p* = 0.987, while it significantly and negatively predicted school volunteering, *B* = −0.08, *p* = 0.035, which in turn positively predicted self-esteem, *B* = 0.02, *p* < 0.001. Bootstrapping results revealed the significant indirect effect of EFUI on self-esteem through school volunteering, *B* = −0.00, bias-corrected 95% CI [−0.005, −0.000]. The overall model *R*^2^ was significant, *F* = 25.50, *p* < 0.001, explaining 6% of the variance in self-esteem.

### 7.3. Auxiliary Analysis

To rule out the possibility that the significant results were due to some spurious associations caused by the covariates, we conducted the analyses without the above covariates and the results remained substantially consistent. Furthermore, we also removed one of the EFUI items from the analyses (i.e., bought/gave virtual goods (e.g., birthday gifts)), which may be perceived as an SFUI item. All model results remained unchanged.

## 8. Discussion

Guided by the aforementioned two lines of inquiry, our study investigated the associations of social function use intensity (SFUI) and entertainment function use intensity (EFUI) with the life satisfaction and self-esteem of vocational college students in China, and examined the potential mediating roles of general prosocial behavior and school volunteering in the interpersonal path of SFUI and the intrapersonal path of EFUI to life satisfaction and self-esteem.

### 8.1. Positive Associations of SFUI with Adolescent Life Satisfaction and Self-Esteem, and Mediating Roles of General Prosocial Behavior and School Volunteering

Our data fully supported the hypothesis of the positive associations of SFUI with adolescent life satisfaction and self-esteem through two interpersonal pathways mediated by general prosocial behavior and school volunteering. Our findings corroborated previous research that demonstrated the extended warming effect of impression management on SNS in enhancing real-life prosocial behavior [55]. Our results were also in line with extant research that points to the positive associations of SNS use with adolescent life satisfaction and self-esteem [31,32,36]. It is noteworthy that when general prosocial behavior and school volunteering were included in our mediation analysis, remarkable reductions in the direct effects of SFUI on life satisfaction and self-esteem were observed, illustrating the crucial mediating roles of general prosocial behavior and school volunteering in the interpersonal pathways from SFUI to adolescent life satisfaction and self-esteem.

### 8.2. Negative Associations of EFUI with Adolescent Life Satisfaction and Self-Esteem, and Mediating Roles of General Prosocial Behavior and School Volunteering

On the other hand, our data partially supported the hypothesis regarding the negative associations of EFUI with adolescent life satisfaction and self-esteem through two intrapersonal pathways mediated by general prosocial behavior and school volunteering. Our findings on the overall negative associations of EFUI with adolescent life satisfaction and self-esteem via school volunteering were largely consistent with a previous study on a sample of young SNS users in China [39], which indicated that time spent watching short-form videos on SNS negatively predicted well-being indicators including life satisfaction and positive affect. What is unexpected from our findings is that EFUI did not predict general prosocial behavior, but significantly and negatively predicted school volunteering and in turn positively predicted life satisfaction and self-esteem. This can be explained by the nature of EFUI activities and the differences between school volunteering and general prosocial behavior. While EFUI may reduce the time that adolescents invest in empathy building in real-life social interactions, the negative impact is likely to be stronger on school volunteering than on general prosocial behavior. The reason is that typical EFUI activities such as watching livestreams, shopping, and gaming involve free choices and have fewer constraints. Hence, they are in stark contrast to school volunteering, which is highly structured, formal, and institutionalized. On the contrary, general prosocial behavior (e.g., giving directions to a stranger) is usually self-initiated and freely performed in everyday life, and therefore is less restrictive and less controlled as compared to school volunteering. In other words, there are more similarities between general prosocial behavior and EFUI activities. Moreover, given that many online games involve cooperation and collaboration tasks, players may extend their online helping acts into real life, which ultimately offsets the negative associations of EFUI with general prosocial behavior. Taken together, the distinction between general prosocial behavior and school volunteering in terms of their nature may explain the differential associations of EFUI activities with them.

## 9. Conclusions

Based on a large student sample from 10 vocational colleges in four regions of China, our research has shed light on how the use of different SNS functions relates to prosocial behavior and well-being among Chinese adolescents. We found that SFUI positively predicted adolescent life satisfaction and self-esteem, through two separate interpersonal pathways mediated by general prosocial behavior and school volunteering. Meanwhile, EFUI negatively predicted adolescent life satisfaction and self-esteem, through an intrapersonal pathway mediated by school volunteering. Our findings highlighted the differential associations of SNS function uses with adolescent well-being and unpacked the underlying mechanisms of prosocial behavior. We also revealed that SNS use can be a double-edged sword in enhancing adolescent prosocial behavior and well-being. Although making social connections remains the primary purpose of SNS use globally, there is a clear trend toward entertainment use in many non-Western countries, especially China, a pioneer in livestream shopping and short-form video watching [2]. This is particularly worrying, given the negative impacts of EFUI on adolescent prosocial behavior and well-being as evidenced in our findings, and considering the huge number of adolescents who will be using SNS for entertainment purposes, if they have not been using it already. Therefore, educators should promote media and mental health literacy to adolescents, so as to better equip them for the opportunities and challenges that may arise from SNS use. At the same time, parents and teachers should ascertain clear guidance for SNS use, thereby tapping into the benefits of online activities without sacrificing the well-being of adolescents. Furthermore, charitable organizations may explore the use of SNS to engage the young generation in volunteering. One example is Oxfam, an international charity for poverty. Specifically, besides using SNS to advertise volunteering opportunities, Oxfam also recruits volunteers as young as 14 years old to manage its social media channels, with the main tasks of creating content, driving online engagement, and promoting charity campaigns across different SNS [74]. We encourage researchers in the fields of social, developmental, and educational psychology to further investigate the associations of SNS use with adolescent well-being by identifying the antecedents and examining the short- and long-term consequences of SNS use. Building on our empirical evidence, social work practitioners should be able to develop SNS-based intervention programs, and assess their effectiveness on adolescent well-being.

Nonetheless, there are several caveats in our research that warrant attention. Firstly, due to the self-reporting nature of the survey, participants’ responses might be subject to social desirability, introspective ability, and recall bias. Future research may consider using the automated tracking approach [75] to validate adolescents’ self-report data. Secondly, given the cross-sectional nature of the study, current data cannot provide sufficient support to establish causal relationships between variables. Further investigation may include multi-wave longitudinal or experimental studies to increase the robustness of findings and facilitate the understanding of the relationships between specific variables. Thirdly, although a global measure of self-esteem is widely used in psychological research, it only depicts an overview of one’s belief of self-worth, without taking into account the unique self-evaluation at specific domains. Hence, subsequent research should assess participants’ self-esteem across different domains, such as the social, academic, athletic, and appearance domains, which might be particularly central to the developmental stage of adolescence [76]. Fourthly, in spite of the fact that the Satisfaction With Life Scale has been the predominant measure of life satisfaction in the past three decades, future research may consider using alternative measures such as the Riverside Life Satisfaction Scale (RLSS) [77] to capture a broader concept of life satisfaction with multiple indirect indicators and a balance of positive- and negative-worded items. Fifthly, the student sample in the current study was exclusively drawn from vocational colleges, which represented an alternative educational path commonly taken by adolescents with migrant family backgrounds, lower socioeconomic status (SES), and fewer opportunities in the formal academic track, as compared to students at mainstream academic schools [68,78]. Even though vocational colleges account for 40% of the total number of students registered in the higher education category in China [66], psychological research on this unique population is relatively scarce. As a result, our work has not only contributed to the understanding of the relationships among SNS use, prosocial behavior, and well-being in this under-researched community, but also laid an important groundwork for further investigation in other populations. Future research should extend our empirical inquiry to academic schools for the more affluent adolescents, so as to further explore the moderating roles of SES in the links. Sixthly, despite the large sample size of the present study, the participants were recruited through convenience sampling and were limited to four regions of China. To increase the generalizability of findings across cultures, future research may recruit a representative sample of adolescents from other countries with not only different cultural backgrounds, socioeconomic conditions, and political environments, but also a fundamentally distinctive SNS ecosystem [2]. Lastly, the present study focused on adolescents only, as they are the most active SNS users and are most susceptible to the effects of SNS use. In view of the increasing popularity and influence of SNS across age groups, future studies may probe into other cohorts, especially older adults [79,80,81]. One potential direction is to examine the roles of technophobia in the link between the types of SNS use and elderly well-being [80,82,83]. Notwithstanding the above caveats, our study has made a major and novel contribution to the under-researched area of SNS use, prosocial behavior, and well-being among Chinese adolescents. Our fresh insights should help advance the understanding of this important topic and inform future lines of research.

## Figures and Tables

**Figure 1 ijerph-20-00100-f001:**
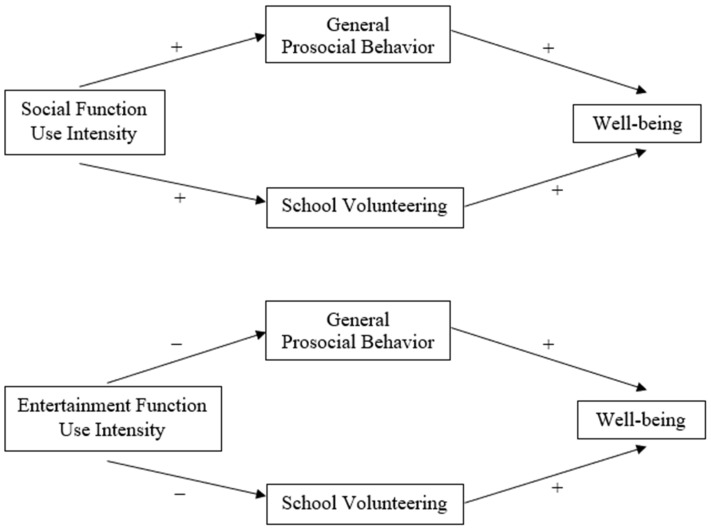
Interpersonal vs. Intrapersonal Mediating Pathways of Prosocial Behavior by Social Function Use Intensity vs. Entertainment Function Use Intensity to Well-Being.

**Figure 2 ijerph-20-00100-f002:**
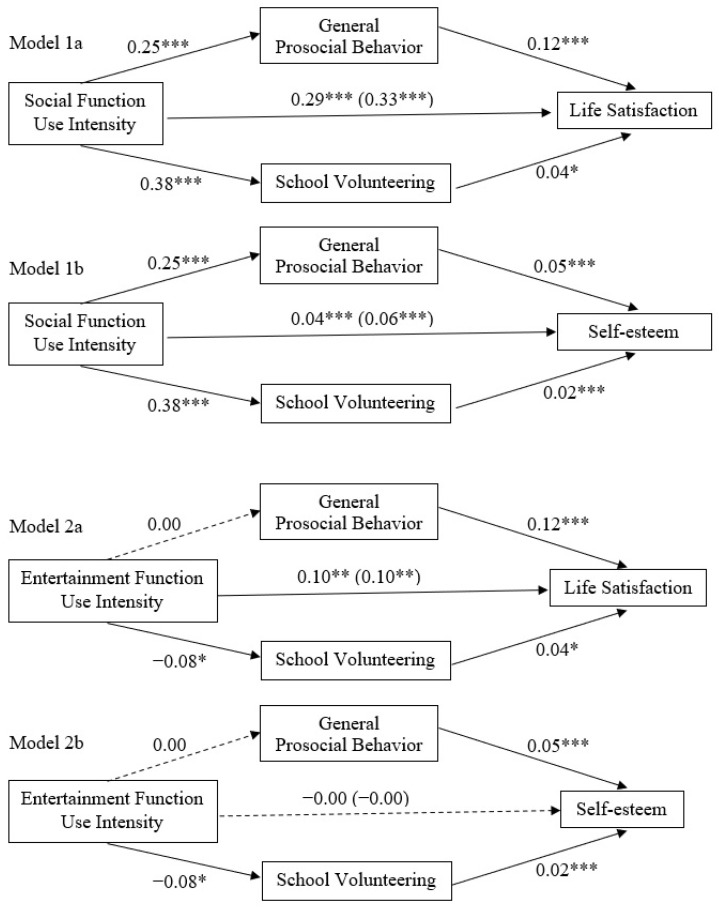
Unstandardized coefficients of the mediation model (controlling for other function use intensity, sex, age, smartphone use frequency, and school regions) depict prosocial behavior as a mediator between SNS function use and well-being. Note. *N* = 3452. Coefficients in parentheses illustrate the total effect without the mediators. EFUI was controlled for in Models 1a and 1b while SFUI was controlled for in Models 2a and 2b. Dashed lines represent a nonsignificant relationship (*p* > 0.05). * *p* < 0.05. ** *p* < 0.01. *** *p* < 0.001.

**Table 1 ijerph-20-00100-t001:** Summary of Means, Standard Deviations, and Intercorrelations for the Variables of Interest (*N* = 3452).

		*M* (*SD*)	2	3	4	5	6	7	8	9
1.	Sex ^a^	—	0.12 ***	0.01	0.05 **	0.06 ***	0.00	0.06 **	0.01	0.06 ***
2.	Age	18.21 (2.01)	—	0.01	0.06 **	0.15 ***	0.14 ***	0.02	0.10 **	0.12 ***
3.	SFUI	2.99 (0.70)		0.92	0.63 ***	0.27 ***	0.20 ***	0.25 ***	0.13 ***	0.12 ***
4.	EFUI	3.19 (0.70)			0.72	0.15 ***	0.07 ***	0.20 ***	0.10 ***	0.32 ***
5.	General prosocial behavior	2.69 (0.65)				0.91	0.60 ***	0.16 ***	0.16 ***	0.01
6.	School volunteering	2.65 (1.15)					—	0.13 ***	0.15 ***	0.02
7.	Life satisfaction	4.14 (1.11)						0.90	0.30 ***	0.03 *
8.	Self-esteem	2.71 (0.37)							0.70	0.13 ***
9.	Smartphone use frequency	4.07 (1.00)								—

*Note.*^a^ Male = 1, Female = 2. SFUI = Social function use intensity. EFUI = Entertainment function use intensity. Reliability coefficients are found along the diagonal line. ** p* < 0.05. *** p* < 0.01. **** p* < 0.001.

## Data Availability

The data presented in this study are available on request from the corresponding author.

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
