# Peer review of "From Social Networking Site Use to Subjective Well-Being: The Interpersonal and Intrapersonal Mediating Pathways of Prosocial Behavior among Vocational College Students in China"

_ijerph, 2022, doi:10.3390/ijerph20010100_

Round 1

Reviewer 1 Report

The manuscript “From Social Networking Use to Subjective Well-Being: The Interpersonal and Intrapersonal Mediating Pathways of Prosocial Behavior among Vocational College Students in China” is a relevant study and is of particular importance with the increasing use of social networks and inherent psychological implications in adolescents.

Despite being a study focused on a particular reality in China, both the methodological approach and its interpretation are sound and, at this stage, require very few (if any) adjustments. The results are clearly expressed and  implications are very coherent and thorough. There is, as correctly mentioned by the authors, many other interesting fields to be explored building on these results.

I would, however, like to ask the authors if they would not want to consider including the average response time in the (extensive) questionnaire (50+ items) for the sake of replication.

In addition to this, I also question why did the authors not choose to normalize the Likert scales to just one gradation, instead of using maximum values of 4, 5 and 7, depending on the different measures. This could help to soften some of the participants' cognitive burden.

A final note to question why, in the footnote of Table 1, 'Male – 1, Female – 2.' is mentioned when there is no information available for these data.

Reviewer 2 Report

Researchers present a study on the effects of social function and entertainment use of social networks on adolescents' life satisfaction and self-esteem. This research is very interesting because of the current subject matter of the study. Nevertheless, there are some changes that authors should make before publication of the article:

1. First of all, the novelty of the study compared to previous studies in the field should appear before the research objectives. Authors should highlight the contributions that this research aims to make and that have not been addressed in other studies.

2. Secondly, at the end of the introduction (not in a separate section), the general objective of the introduction should be clearly and directly stated. Then, the specific objectives underlying this general objective should be described. Again, these objectives should be written clearly and concisely.

3. Thirdly, the method section should be restructured. Thus, authors should begin this section by referring to the research design: what approach is used, is it a non-experimental design, what type of design, and what is the research design?

4. Fourthly, and following on from the method section, the participants and procedure sections should be divided. Thus, a section referring to the characteristics of the participants and another section referring to the procedure followed during the research should be included.

5.- The sub-section "Measures" should be replaced by "Instruments".

6.- The section on results should be structured in sub-sections, each of them referring to a specific objective.

7.- The most critical point is the section on discussions. First of all, the authors should divide between the discussions and the conclusions, as they are not the same thing. At present, the article has no conclusions.

8.- As for the discussions, they should be structured according to the specific objectives and the findings. In this sense, each of the results should be justified with other research that explains the cause of these. In addition, as the authors rightly do, reference should be made to other research that is in line with the results.

9.- The conclusions section should include: a general conclusion, specific conclusions that respond to the specific objectives, the limitations of the study, future lines of research and the practical implications of the study.
